# The Control of Fusarium Root Rot and Development of Coastal Pine (*Pinus thunbergii* Parl.) Seedlings in a Container Nursery by Use of *Bacillus licheniformis* MH48

**Sang-Jae Won [1], Vantha Choub [1], Jun-Hyeok Kwon [1], Dong-Hyun Kim [2] and Young-Sang Ahn [1,***

[1] Division of Forest Resources, Chonnam National University, Gwangju 61186, Korea; lazyno@naver.com (S.-J.W.); vanthachoub@gmail.com (V.C.); wg6102@naver.com (J.-H.K.)
[2] Department of Fire Safety Engineering, Jeonju University, Jeollabuk-do 55069, Korea; 72donghyunkim@jj.ac.kr
* Correspondence: ysahn@jnu.ac.kr; Tel.: +82-62-530-2081; Fax: +82-62-530-2089

**Abstract:** This study investigated the control of Fusarium root rot and development of coastal pine (*Pinus thunbergii*) seedlings in a container nursery by using *Bacillus licheniformis* MH48. High-quality seedlings without infectious diseases cause vigorous growth. Fusarium root rot caused by *Fusarium oxysporum* is responsible for serious damage to coastal pine seedlings in nurseries. *B. licheniformis* MH48 produced enzymes that degraded the fungal cell walls, such as chitinase and β-1,3-glucanase. These lytic enzymes exhibited destructive activity toward *F. oxysporum* hyphae, which were found to play key roles in the suppression of root rot. In addition, *B. licheniformis* MH48 increased the nitrogen and phosphorus in soils via fixed atmospheric nitrogen and solubilized inorganic phosphate. *B. licheniformis* MH48 produced the phytohormone auxin, which stimulated seedling root development, resulting in increased nutrient uptake in seedlings. Both the bacterial inoculation and the chemical fertilizer treatments significantly increased seedling growth and biomass, and the bacterial inoculation had a greater effect on seedling development. Based on the results from this study, *B. licheniformis* MH48 showed potential as a biological agent against Fusarium root rot and as a promoter of growth and development of *Pinus thunbergii* seedlings.

**Keywords:** antagonistic bacteria; lytic enzyme; root rot pathogen; auxin; seedling development; container nursery

## 1. Introduction

Coastal pine (*Pinus thunbergii* Parl.), which can tolerate drought and salt stress, grows well in sandy soil and in full sun and is an important tree in the coastal forests that are widespread in Northeastern Asia [1–4]. Several studies on understanding the process and mechanisms of establishing coastal pine forests, as well as reforestation projects, have been initiated, with the goal of recovering and maintaining coastal pine forests [1–5]. Coastal forests growing along the shoreline can help reduce the devastating impact of a tsunami and storm surge by decreasing their wave energies [6]. In addition, they provide a variety of services for coastal ecology and societies, including wildlife habitat [7], water-quality control [8], and shelter for people [9].

In a recent forest restoration project, container seedlings have mostly replaced bare-root seedlings [10–12]. Container seedlings promote early growth after outplanting by reducing shock and providing better seedling health and a better survival rate [11]. However, the production of seedlings

using containers can cause defects, such as nutritional imbalances [13]. In addition, excessive use of chemical fertilizers in forest nursery production damages soil microorganisms, affects the fertility of soil, decreases plant growth, and pollutes the environment [14–16]. In particular, nitrogen fertilizer tends to substantially increase the proportion of fungal pathogenic genera, thus worsening plant health [17]. The container seedlings are often exposed to persistent pathogenic attacks, particularly those inciting root rot, at primary stages of seedling establishment. Root rot in container-grown conifers such as spruce and pine is a major problem [18–20]. During a nursery survey in this study, root rot incited by *Fusarium oxysporum* was found in coastal pine seedlings, and many seedlings wilted and died (Figure 1). Given the highly devastating nature of root rot pathogens, effective disease management is essential to raise healthy forest seedlings for the successful implementation of reforestation and afforestation. Fungicide use has been adopted by nursery growers to reduce the incidence of root rot, but the disease continues to pose a serious threat to seedlings [21]. In addition, many chemical products, including fertilizers and fungicides, deplete nonrenewable resources and pose hazards to humans and the environment [22,23]. Therefore, interest in environmentally friendly practices in forest seedling production systems has recently focused on promoting seedling development.

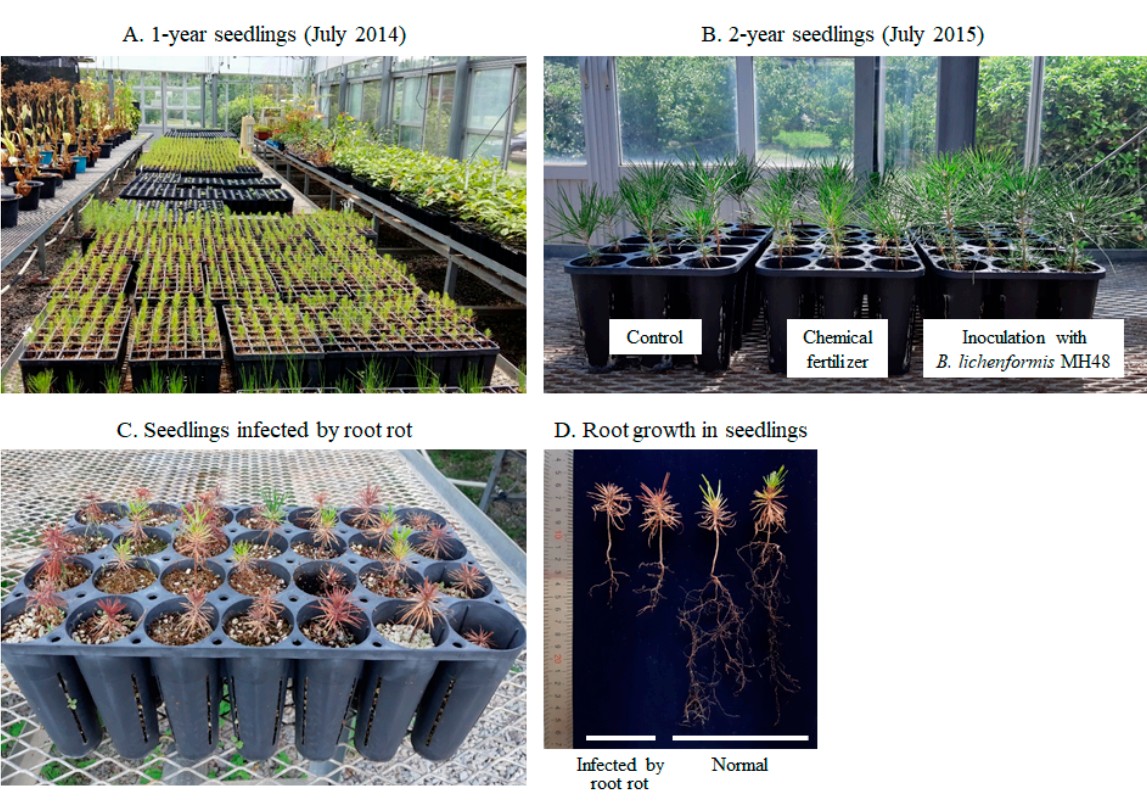

**Figure 1.** Study area (**A**) and seedling growth (**B**) of coastal pine in an experimental greenhouse at the forest nursery. Seedlings infected with Fusarium root rot wilted and died (**C**), and root growth was reduced (**D**).

Plant growth-promoting rhizobacteria (PGPR) are beneficial native soil bacteria that are an environmentally sound way of increasing plant yields by facilitating plant growth through either a direct or an indirect mechanism [24–40]. The direct promotion of plant growth by PGPR provides the plant with phytohormones [24–26], which facilitate the uptake of nutrients from the soils [24,25,27,28] and help establish a symbiosis with rhizobia or mycorrhiza [29]. The indirect promotion of plant growth occurs when PGPR lessen or prevent the deleterious effects of pathogenic organisms [30–37], nematodes, and insects [38–40] by producing antagonistic substances, such as hydrolytic enzymes, including chitinases, glucanases, proteases, and lipases. The *Bacillus* species is one of the most important microorganisms in PGPR and is isolated from rhizosphere soil [25,26,34,37,39,41].

The benefits of *Bacillus licheniformis* strains include the production of plant hormones, such as auxin and gibberellin, nitrogen fixation, and the solubilization of inorganic phosphate, which lead to increased forest seedling growth and yield [24–26]. In addition, *B. licheniformis* strains are bioenhancers that stimulate fine-root development and promote nutrient uptake in forest seedlings [24]. However, to date, the effects of lytic enzymes derived from *B. licheniformis* strains on fungal pathogens remain poorly understood.

Forestry has studied many bacterial species that act as PGPR, described in the literature as being successful in the biological control of diseases [18–20,42] and for improving plant growth [24–26,29,43–46]. However, studies of PGPR being used simultaneously for biological disease control and tree-growth promotion are rarely described in forestry. The increasing demand for forestry production with a significant reduction of synthetic chemical fertilizer and fungicide use is a big challenge currently. The rapid early growth of healthy, infection-free forest seedlings is closely related to the success of plantation establishment [47] because after planting, when they compete for resources with weeds, large seedlings perform better than small seedlings [48,49]. To produce high-quality coastal pine seedlings for the construction of coastal pine forests, it is very important to understand the effects of PGPR on the development of these seedlings in a container nursery. The objective of this study was to investigate the control of Fusarium root rot and growth promotion of coastal pine seedlings in a container forest nursery by using *B. licheniformis* MH48.

## 2. Materials and Methods

### 2.1. Antagonistic Activity of B. licheniformis MH48 against F. oxysporum

The bacterial strains of *B. licheniformis* MH48 were isolated from rhizosphere soil in a coastal area of Korea [25,26,39,41]. This bacterial strain was obtained from the laboratory of soil microbiology, Chonnam National University, Korea. During a nursery experiment in this study, coastal pine seedlings had rotten roots and died (Figure 1C). According to the Korean Agriculture Culture Collection (KACC; Suwon, Korea), *F. oxysporum* was found in dead seedlings of coastal pines. Fungal pathogen *F. oxysporum* was purchased from KACC for antagonistic activity experiments with *B. licheniformis* MH48.

For the control experiment of Fusarium root rot from *F. oxysporum* with lytic enzymes of *B. licheniformis* MH48, the latter was cultured in 1 L of broth medium and incubated at 30 °C for 7 days. The composition (per liter) for the broth medium was urea $((NH_2)_2CO)$ (1.7 g), potassium phosphate monobasic $(KH_2PO_4)$ (0.4 g), potassium chloride $(KCl)$ (0.08 g), organic compost (1.0 g), and sugar (2.2 g). Growth of *B. licheniformis* MH48 was examined for the required cell density ($10^7$ colony forming units (CFU) $mL^{-1}$) and checked for contamination before application [25].

The antagonistic activity of *B. licheniformis* MH48 against *F. oxysporum* was measured by the dual culture method. A potato dextrose agar (PDA) medium was prepared, *B. licheniformis* MH48 was streaked onto one side of each agar plate, and a fungal agar plug of 5 mm in diameter was made using a sterile cork borer and placed on the other side of the inoculated plates. A plate inoculated with the fungal pathogen alone was used as the control. Three replicates of each plate were incubated at 30 °C for 7 days, and the proportion of *B. licheniformis* MH48 inhibiting the growth of the fungal pathogen *F. oxysporum* was calculated using the formula [50]: inhibition (%) = $((R_1 - R_2)/R_1) \times 100$, where $R_1$ is the radial growth of *F. oxysporum* in the control plate, and $R_2$ is the radial growth of *F. oxysporum* in the dual culture plate.

### 2.2. Production of the Defense-Related Lytic Enzymes by B. licheniformis MH48

To examine chitinase and β-1,3-glucanase activities, *B. licheniformis* MH48 was cultured on the medium at 30 °C for 7 days, and the bacterial supernatant was collected daily. To assay the chitinase activity, a reaction mixture consisting of 50 μL of bacterial culture, 450 μL of 50 mM sodium acetate buffer (pH 5.0), and 500 μL of a 0.5% colloidal chitin solution was incubated at 37 °C for 1 h [51]. The reaction was terminated by adding 200 μL of 1 N NaOH and centrifuged at 12,000 rpm for 10 min

at 4 °C. The supernatant (500 μL) was mixed with 1 mL of Schales' reagent and boiled for 15 min at 100 °C in a water bath. Absorbance was measured at 420 nm by a UV spectrophotometer. One unit of chitinase activity was defined as the reducing activity that releases 1 μmol of *N*-acetylglucosamine per hour at 37 °C.

To measure the β-1,3-glucanase activity, a reaction mixture containing 50 μL of bacterial culture, 50 μL of laminarin (10 mg mL$^{-1}$), and 400 μL of 50 mM of a sodium acetate buffer (pH 5.0) was incubated at 37 °C for 1 h [52]. The reaction was stopped by adding 1.5 mL of dinitrosalicylic acid (DNS) reagent and boiling in a water bath for 5 min. Absorbance at 550 nm was used to measure the concentration of reducing sugars. One unit of β-1,3-glucanase activity was defined as the amount of enzyme that catalyzes the release of 1 μmol of glucose per hour at 37 °C.

### 2.3. Effect of Lytic Enzymes Produced from B. licheniformis MH48 on the Mycelial Morphology of F. oxysporum

To examine how much the lytic enzymes, including chitinase and β-1,3-glucanase, from *B. licheniformis* MH48 inhibited *F. oxysporum* mycelia growth, 1 mL of *F. oxysporum* culture was grown on a potato dextrose broth (PDB) medium at 30 °C for 3 days. *F. oxysporum* mycelia were observed under a light microscope (Olympus BX41TF, Tokyo, Japan). All observations of morphological mycelia were performed in triplicate.

### 2.4. Plant Material and Experimental Conditions

The coastal pine seeds used in this study were obtained in December 2013 from the National Forest Seed Variety Center of Korea and stored at 4 °C in a cool refrigerator with a constant relative humidity of 40%. To interrupt their dormancy, these seeds were dipped in running tap water for 7 days. Seeds were sown at 1 cm depth of growth media in each container (volume 500 mL). The seedling tray was 25 cm wide, 41 cm long, and 16 cm high, and included 15 cells with a 7.5 cm diameter and 16 cm height (Figure 1B). Seedling emerged in March 2014, approximately 1 month after sowing seeds in February 2018. The peat was turfy peat with a medium structure, and the perlite was coarse (peat/perlite = 5:1). After the peat and perlite were mixed, the growth medium was sieved to 5 mm, and the containers were manually filled with approximately 500 mL substrate.

The experiment was carried out with three replications in a greenhouse with an automatic spray irrigation system at the forest nursery at Chonnam National University (approximately 35°17′ N latitude, 126°90′ E longitude) (Figure 1). The experiment was conducted in a greenhouse, with the temperature kept at 20–25 °C all day long using a heating and cooling system. The light condition was natural. The following three treatments were used: control (without fertilizer or bacteria), a chemical fertilizer, and *B. licheniformis* MH48 inoculation. Each treatment involved 45 seedlings (15 seedlings for each replication), and a total of 135 container seedlings were used in the experiment. During the 12 months of growth in the experimental period from March 2014 to February 2015, normal management practices were carried out, but no fertilization or bacterial inoculation was applied. From March 2015 to February 2016, the container seedlings were treated with chemical fertilizer or *B. licheniformis* MH48 inoculation at two-week intervals. The bacterial inoculation (10 L of *B. licheniformis* MH48 culture) and the chemical fertilizer (solved urea (17.0 g), potassium phosphate monobasic (40.0 g), and potassium chloride (8.0 g) in 10 L of water) were added to 10 L of water and were applied to the soil adjacent to the roots. The fertilizer application rate was based on the recommended basal chemical fertilizer application rate for coniferous seedlings in the nursery of the Korea Forest Service [53].

### 2.5. Growth Medium, Plant Sampling, and Measurement

In February 2016, the growth media were mixed in each replication, and samples (*n* = 9) were taken from each replication and oven-dried at 105 °C for 24 h.

After we carefully washed the root systems to eliminate all media, the seedling growth parameters were measured, including shoot and root lengths, root collar diameter, and seedling biomass (shoot and root dry mass) (*n* = 135). The biomass was measured after drying the separated shoots and roots at 105 °C for 24 h. The shoots and roots were separated at the boundary of the uppermost part of the root. Additionally, based on the five selected seedlings of average growth that were measured for each treatment (*n* = 15), measurements were taken of the total nitrogen and total phosphorus of the seedlings (*n* = 45). For the calculation of the nutrient content of the seedlings, the following formula was used [25,26]: nutrient content (mg) = ((dry weight (g) × nutrient concentration (%)).

The total nitrogen content of the growth media was calculated by the Kjeldahl method [54] after wet digestion with $H_2SO_4$. The total nitrogen content of the seedlings was analyzed using an elemental analyzer (Variomax CN Analyzer, Elementar Analysensysteme GmbH, Germany) with a thermal conductivity detector (TCD) after combustion at 1200 °C with nitrogen and helium. After digestion in nitric acid in a microwave oven (MARS Xpress, CEM Corporation, Matthews, NC, USA), the total phosphorus content in the growth media and seedling samples was measured using an ICP-OES (Optima 8300, PerkinElmer, Waltham, MA, USA).

*2.6. Statistical Analysis*

The data were subjected to an analysis of variance (ANOVA) using the SPSS statistics program (version 21; Chicago, IL, USA). The mean values were compared using a Least Significant Difference (LSD) test at a significance level of $p < 0.05$, and the results are reported as the mean ± standard deviation. Variables of seedling nutrient concentration, shoot length, root length, total height, and root dry mass were log-transformed to stabilize the variance.

## 3. Results

*3.1. Control of B. licheniformis MH48 against Fusarium Root Rot Disease*

3.1.1. Production of Lytic Enzymes by *B. licheniformis* MH48

The production of lytic enzymes, such as chitinase and β-1,3-glucanase activities, was examined from the bacterial culture of *B. licheniformis* MH48 (Figure 2). The chitinase activity gradually increased for five days, eventually reaching a maximum value of 428.30 units $mL^{-1}$. Thereafter, the chitinase activity rapidly decreased (Figure 2A). The β-1,3-glucanase activity slowly increased for three days, eventually reaching a maximum value of 15.25 units $mL^{-1}$ of the growth of *F. oxysporum*.

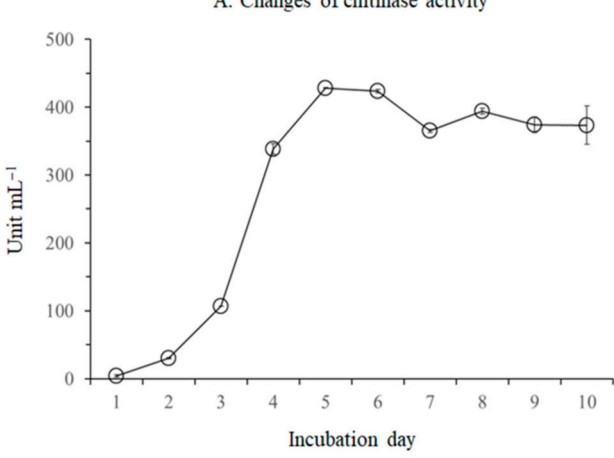

**Figure 2.** *Cont.*

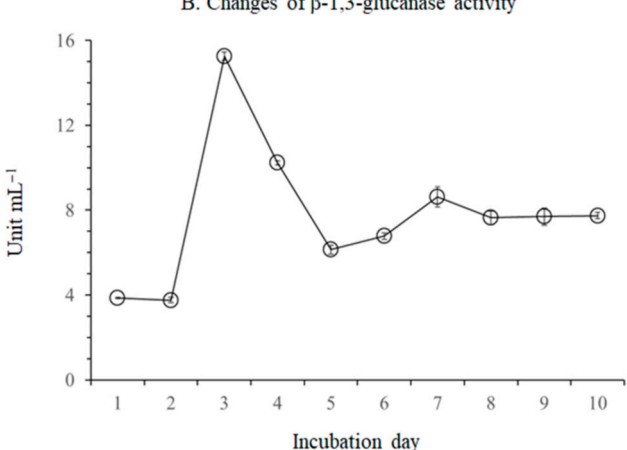

**Figure 2.** Changes in chitinase activity (**A**) and β-1,3-glucanase activity (**B**) in the culture of *B. licheniformis* MH48. Error bars represent the standard deviation of three replications.

The hyphae of *F. oxysporum* incubated without *B. licheniformis* MH48 as the control showed normal morphology under the light microscope (Figure 3A). However, the hyphae of four foliar fungal pathogens incubated with *B. licheniformis* MH48 showed mycelial abnormalities, such as degradation, deformation, and lysis (Figure 3B).

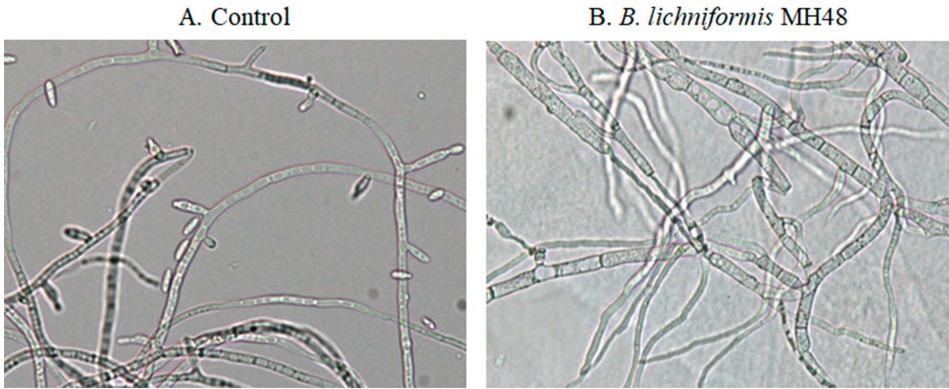

**Figure 3.** Effect of chitinase and β-1,3-glucanase produced by *B. licheniformis* MH48 on hyphal deformation of *F. oxysporum* (**B**) and the control (**A**) under light microscope.

*3.2. Effects of Inoculation with B. licheniformis MH48 on the Development of Coastal Pine Seedlings*

3.2.1. Nutrient Contents of the Growing Media

The total nitrogen content was between 0.15 and 0.21 g $kg^{-1}$, and for the control, chemical fertilizer, and inoculation with *B. licheniformis* MH48 treatments, the total phosphorus content ranged from 0.05 to 0.06 g $kg^{-1}$ in the growth media (Table 1). The total nitrogen contents of the growth media were significantly lower ($p < 0.05$) after the treatment with bacterial inoculation when compared to both chemical fertilizer and control treatments (Table 1). Bacterial inoculation and chemical fertilizer treatments had no significant effect ($p > 0.05$) on the total phosphorus contents of the growth media (Table 1).

**Table 1.** Total nitrogen and total phosphorus contents of growth media and concentrations and contents of total nitrogen and total phosphorus in coastal pine seedlings for the control, chemical fertilizer, and inoculation with *B. licheniformis* MH48 treatments.

| Treatment | Growth Media Nutrient Content (g kg$^{-1}$) | | Seedling Nutrient Concentration (%) | | Seedling Nutrient Content (mg) | |
|---|---|---|---|---|---|---|
| | Total Nitrogen | Total Phosphorus | Total Nitrogen | Total Phosphorus | Total Nitrogen | Total Phosphorus |
| Control | 0.21 ± 0.04 a | 0.05 ± 0.02 | 0.69 ± 0.15 c | 0.15 ± 0.05 b | 17.04 ± 5.07 b | 3.67 ± 1.36 b |
| Chemical fertilizer | 0.20 ± 0.02 a | 0.05 ± 0.01 | 1.41 ± 0.16 a | 0.23 ± 0.03 a | 46.62 ± 13.03 a | 7.65 ± 2.38 a |
| Bacterial inoculation | 0.15 ± 0.01 b | 0.06 ± 0.01 | 1.07 ± 0.15 b | 0.23 ± 0.03 a | 42.06 ± 12.21 a | 9.24 ± 2.81 a |

Note: Means within each column with the same letter are not significantly different from each other according to the least significant difference test (LSD) at the $p < 0.05$ level.

### 3.2.2. Seedling Growth and Biomass

The total height of the coastal pine seedlings for the three treatments ranged from 29.07 to 37.04 cm, and the root-collar diameter ranged from 3.38 to 4.01 mm (Table 2). The total height and the root-collar diameter of the coastal pine seedlings inoculated with *B. licheniformis* MH48 were significantly higher ($p < 0.05$) than in the chemical fertilizer or control (Table 2). In the chemical fertilizer treatment, the root-collar diameter was significantly higher than in the control treatment, whereas total height was not significantly different from that of the control (Table 2).

**Table 2.** Growth and biomass production of coastal pine seedlings for the control, chemical fertilizer, and inoculation with *B. licheniformis* MH48 treatments.

| Treatment | Seedling Growth | | | | Seedling Biomass | | |
|---|---|---|---|---|---|---|---|
| | Shoot Length (cm) | Root Length (cm) | Total Height (cm) | Root Collar Diameter (mm) | Shoot Dry Mass (g) | Root Dry Mass (g) | Total Dry Mass (g) |
| Control | 15.23 ± 3.23 c | 13.81 ± 2.15 b | 29.04 ± 4.18 b | 3.38 ± 0.32 c | 1.99 ± 0.34 c | 0.45 ± 0.14 b | 2.44 ± 0.45 c |
| Chemical fertilizer | 19.10 ± 2.38 b | 11.73 ± 1.96 c | 30.83 ± 2.36 b | 3.62 ± 0.34 b | 2.85 ± 0.48 b | 0.47 ± 0.12 b | 3.32 ± 0.55 b |
| Bacterial inoculation | 21.10 ± 3.86 a | 15.94 ± 1.96 a | 37.04 ± 4.85 a | 4.01 ± 0.48 a | 3.40 ± 0.56 a | 0.57 ± 0.12 a | 3.97 ± 0.64 a |

Note: Means within each column with the same letter are not significantly different from each other according to the least significant difference test (LSD) at the $p < 0.05$ level.

Compared to the control, the bacterial inoculation and chemical fertilizer treatments had a more significant influence on the shoot, root, and total dry mass of the coastal pine seedlings (Table 2). The seedling biomass production of inoculation with *B. licheniformis* MH48 was greater than of chemical fertilizer, which was greater than that of control treatments. In particular, compared to the biomass of the control seedlings, the shoot, root, and total dry mass of the coastal pine seedlings grown in the bacterial inoculation were higher by 171%, 127%, and 162%, respectively. Clearly, the superior nutrients afforded by bacterial inoculation and chemical fertilizer generated significant seedling biomass, and the bacterial inoculation treatment was especially favorable to the coastal pine seedling biomass (Table 2).

### 3.2.3. Seedling Nutrient Concentration and Content

The concentration and content of total nitrogen and total phosphorus in the coastal pine seedlings treated with chemical fertilizers or inoculated with *B. licheniformis* MH48 were significantly higher than those of the control (Table 1). Except for the total nitrogen concentration in the seedlings, the nutrient concentrations and contents of the seedlings did not significantly differ between the chemical fertilizer and bacterial inoculation (Table 1). The total nitrogen and total phosphorus concentrations of the seedlings in chemical fertilizers and bacterial inoculation were 155%–204% and 153% higher than those in the control, respectively, and the total nitrogen and total phosphorus content by seedlings which received chemical fertilizers and bacterial inoculation were 247%–274% and 208%–252% greater than those in the control, respectively (Table 1).

The highest growth and biomass (37.04 cm and 3.96 mg, respectively) of the seedlings were observed for the bacterial inoculation, which had a high nutrient content, while lower values (29.07 cm and 2.44 mg, respectively) were observed in the control, which had the lowest nutrient content (Tables 1 and 2).

## 4. Discussion

In the chemical fertilizer treatment, several seedlings wilted and showed low growth (Figure 1B). This finding suggests that chemical fertilizers, especially nitrogen fertilizers, tend to substantially increase the proportion of fungal pathogens and harm plant health [17]. However, root rot of coastal pine seedlings by *F. oxysporum* was not shown in the bacterial inoculation treatment (Figure 1B) because *B. licheniformis* MH48 secretes chitinase and β-1,3-glucanase that can degrade chitin and β-1,3-glucan, respectively (Figure 2). Chitin and β-1,3-glucan are major components of fungal cell walls [30]. The hyphae of *F. oxysporum* incubated with *B. licheniformis* MH48 showed mycelial abnormalities, such as degradation, deformation, and lysis (Figure 3B). *B. licheniformis* MH48 showed strong inhibition against target pathogen *F. oxysporum* (Figure 4). Several studies have shown that antagonistic bacteria are an important source of the lytic enzymes that can prevent the dissemination and lower the virulence of fungal pathogens [30–37].

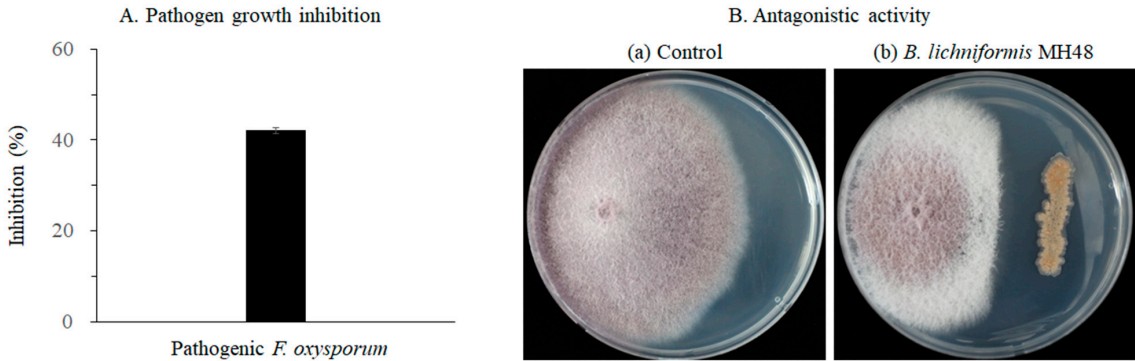

**Figure 4.** Growth inhibition percentage shown in a graph (**A**) and antagonistic activity shown in pictures (**B**) for control (a) and *B. licheniformis* MH48 (b) against *F. oxysporum* by culture method.

Large height and well-developed roots in seedlings are the most essential attributes for vigorous growth following outplanting [10,12]. Nitrogen and phosphorus are essential nutrients for healthy seedling production [25]. *B. licheniformis* MH48 can increase the contents of total nitrogen and total phosphorus in soils via fixed atmospheric nitrogen and solubilizing inorganic phosphate [25,26]. Nevertheless, the total nitrogen and total phosphorus contents in the growth media were not significantly increased by either inoculation with *B. licheniformis* MH48 or the chemical fertilizer (Table 1). In particular, the content of total nitrogen in the growth media after treatment with bacterial inoculation was significantly decreased (Table 1). This finding indicates that the seedlings in both bacterial inoculation and chemical fertilizer treatments absorbed sufficient nutrients from the growth media (Table 1) and increased their seedling growth and biomass (Table 2). Especially, *B. licheniformis* MH48 can produce auxin [25,26], which can stimulate root development (Table 2), resulting in an increased uptake of nutrients, especially total nitrogen, from the growth media (Table 1). Therefore, inoculation with *B. licheniformis* MH48 may be superior to chemical fertilizer for the growth and yield of container seedlings. Our results indicated that *B. licheniformis* MH48 prevented the dissemination of *F. oxysporum*, lowered the virulence of *F. oxysporum*, and obviously promoted the growth of coastal pine seedlings. Therefore, *B. licheniformis* MH48 is an ideal choice for producing seedlings in a forest nursery.

## 5. Conclusions

High-quality seedlings, without infection from diseases and with well-developed shoots and roots, are better able to survive extended environmental stresses and produce vigorous growth after outplanting. The incidence of *F. oxysporum* growth was significantly reduced by *B. licheniformis* MH48 (Figures 3 and 4), apparently because its enzymes break fungal cell walls (Figure 2). In addition, both the inoculation with *B. licheniformis* MH48 and the use of a chemical fertilizer significantly increased the growth and biomass of coastal pine seedlings, but bacterial inoculation was superior to chemical fertilizer for container seedlings (Table 2). *B. licheniformis* MH48 shows potential for reducing the need to use chemical fertilizers and for enabling the best soil and seedling management practices to achieve more sustainable forestry. Our study provides a valuable technique that can enable nursery managers to control fungal pathogens and grow sustainable high-quality seedlings by the use of *B. licheniformis* MH48. It is an attractive concept to use PGPR to reduce the need for agrochemicals, reduce disease incidence, and improve seedling growth.

**Author Contributions:** This study was designed, directed, and coordinated by Y.-S.A., who provided conceptual and technical guidance for all aspects of the project and wrote the manuscript. S.-J.W. was the principal investigator, contributed to the fieldwork and data analysis, performed the literature search, and helped with the writing of the manuscript. V.C. and J.-H.K. assisted with the fieldwork and analysis of data. D.-H.K. helped in data interpretation and commented on the design of the experiments.

**Funding:** This study was carried out with the support of the R&D program for Forest Science & Technology Projects (No. 2018122B10-1820-AB01) provided by the Korea Forest Service (Korea Forestry Promotion Institute). In addition, this research was supported by the Basic Science Research Program through the National Research Foundation (NRF) of Korea, funded by the Ministry of Education, Science and Technology (No. 2018R1D1A1B07050052). It was also partly supported by the Bio-industry Technology Development Program (111056-05) funded by the Ministry of Agriculture, Food, and Rural Affairs, Republic of Korea.

**Conflicts of Interest:** The authors declare no conflict of interest.

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
