# Peer review of "The Control of Fusarium Root Rot and Development of Coastal Pine (Pinus thunbergii Parl.) Seedlings in a Container Nursery by Use of Bacillus licheniformis MH48"

_forests, doi:10.3390/f10010006_

Round 1
Reviewer 1 Report
In this study, the authors investigate the use of a plant-growth promoting rhizobacteria (PGPR) to replace chemical fertilizers in establishing Coastal Pine seedlings in Nursery situations. They demonstrate an inhibition of the fungal pathogen F. oxysporum by their chosen PGPR (B. licheniformis MH48) in laboratory experiments and an enhancement of growth of coastal pine seedlings. These results are worthwhile as they are of potential interest to nursery managers and forestry professionals. However I feel that the authors do not fully satisfy the requirements of a rigorous scientific paper, mostly through deficiencies in the text rather than the experimental design itself. I list the main points in short below:
- it is not clear what the hypotheses of the study are. What (directional?) responses do the authors expect to the applied treatments? Why do the authors expect this?
- the link between the two halves of the experiment is not clear. The authors show an apparrent inhibition of the fungal pathogen in the lab and an apparrent promotion of tree growth in seedlings. But are the plants in the nursery without the PGPR suffering from root rot? The authors do not explain this nor do they test for F. oxysporum in the nursery. It is necessary to explain this combination more clearly.
- why were the comparisons between a certain innoculation level of B.lincheniformis and specific compositions/amounts of chemical fertilizer used? Are these comparable?
- the authors write that they log-transformed all variables to stabilize the variance. Was this necessary for all variables? The appropriateness of this transformation is dependent on the distribution of the untransformed variables?
- the discussion section repeats information from the introduction and includes basic information which would be better suited introduce the study. The discussion should be used to comment on the specific results of this study with reference to the literature. I think this section needs to be substantially rewritten.
Author Response
Response to Reviewer 1 Comments
We appreciate your valuable time and critical comments. We agree with your suggestions and comments. Therefore, we have revised the manuscript according to your suggestions. Revised sentences are highlighted in yellow in the revised manuscript.
*********************
In this study, the authors investigate the use of a plant-growth promoting rhizobacteria (PGPR) to replace chemical fertilizers in establishing Coastal Pine seedlings in Nursery situations. They demonstrate an inhibition of the fungal pathogen F. oxysporum by their chosen PGPR (B. licheniformis MH48) in laboratory experiments and an enhancement of growth of coastal pine seedlings. These results are worthwhile as they are of potential interest to nursery managers and forestry professionals. However, I feel that the authors do not fully satisfy the requirements of a rigorous scientific paper, mostly through deficiencies in the text rather than the experimental design itself. I list the main points in short below:
- it is not clear what the hypotheses of the study are. What (directional?) responses do the authors expect to the applied treatments? Why do the authors expect this?
► Generally, chemical fertilizers have been used in forest nursery for seedling production. However, chemical fertilizers tend to substantially increase the proportion of fungal pathogens, thus harming plant health. Fusarium root rot caused by F. oxysporum is a major problem in coastal pine seedlings in nurseries. During a nursery survey in this study, root rot incited by F. oxysporum was found in coastal pine seedlings. Several seedlings wilted and showed low growth. However, B. licheniformis MH48 is potential biological agent to suppress root rot because it can produce lytic enzymes that exhibit destructive activity toward F. oxysporum hyphae. In addition, B. licheniformis MH48 can produce phytohormone auxin that can stimulate seedling root development, resulting in increased nutrient uptake and growth in seedlings. The use of B. licheniformis MH48 is expected to be able to reduce the need for chemical fertilizer, reduce disease incidence, and improve seedling growth. This study investigated the control of Fusarium root rot and the growth promotion of coastal pine seedlings in a container forest nursery by using B. licheniformis MH48. We have revised the objective of this study to be clear to readers as follows:
The objective of this study was to investigate the control of Fusarium root rot and growth promotion of coastal pine seedlings in a container forest nursery by using B. licheniformis MH48. (Lines 14-15 and 88-90)
- the link between the two halves of the experiment is not clear. The authors show an apparrent inhibition of the fungal pathogen in the lab and an apparrent promotion of tree growth in seedlings. But are the plants in the nursery without the PGPR suffering from root rot? The authors do not explain this nor do they test for F. oxysporum in the nursery. It is necessary to explain this combination more clearly.
► Root rot of coastal pine seedlings by F. oxysporum causes them to wilt and show low growth during a nursery experiment in this study. We have explained this as follows:
: During a nursery experiment in this study, coastal pine seedlings had rotten roots and died (Figure 1C). According to the Korean Agriculture Culture Collection (KACC; Suwon, Korea), F. oxysporum was found in dead seedlings of coastal pines. (Lines 95-96)
: In the chemical fertilizer treatment, several seedlings wilted and showed low growth (Figure 1B). This finding suggests that chemical fertilizers, especially nitrogen fertilizers, tend to substantially increase the proportion of fungal pathogens and harm plant health. (Lines 265-267)
: In addition, we have revised the objective of this study to be clear to readers as follows:
The objective of this study was to investigate the control of Fusarium root rot and growth promotion of coastal pine seedlings in a container forest nursery by using B. licheniformis MH48. (Lines 14-15 and 88-90)
- why were the comparisons between a certain innoculation level of B. lincheniformis and specific compositions/amounts of chemical fertilizer used? Are these comparable?
► Generally, chemical fertilizers have been used in forest nursery for seedling production. However, chemical fertilizers tend to substantially increase the proportion of fungal pathogens and harm plant health. In this study, we wanted to provide valuable information that can enable nursery managers to control fungal pathogens and grow sustainable high-quality seedlings by using B. licheniformis MH48. The fertilizer application rate was based on the recommended basal chemical fertilizer application rate for coniferous seedlings in the nursery of the Korean Forest Service (Kim et al., 2012). The amount of fertilizer used in B. licheniformis MH48 was applied in the same amount as that of chemical-fertilizer treatment. Therefore, they are comparable between bacterial inoculation and chemical fertilizer (Park et al., 2017a; b). We have explained the amount of fertilizer in B. licheniformis MH48 and chemical fertilizer treatments. (Lines 102-103 and Lines 164-169)
- the authors write that they log-transformed all variables to stabilize the variance. Was this necessary for all variables? The appropriateness of this transformation is dependent on the distribution of the untransformed variables?
► We have revised the objective of this study to be clear to readers as follows:
The data were subjected to an analysis of variance (ANOVA) using the SPSS statistics program (version 21; Chicago, IL, USA). The mean values were compared using a Least Significant Difference (LSD) test at a significance level of p < 0.05, and the results are reported as the mean ± standard deviation. Especially, variables of seedling nutrient concentration, shoot length, root length, total height, and root dry mass were log-transformed to stabilize the variance. (Lines 192-196)
- the discussion section repeats information from the introduction and includes basic information which would be better suited introduce the study. The discussion should be used to comment on the specific results of this study with reference to the literature. I think this section needs to be substantially rewritten.
► We have avoided the repetition about the basic information in the Discussion section. In addition, we have rewritten the Discussion section based on specific results of this study with reference to the literature. (Lines 265-296)

Reviewer 2 Report
Great work. A well-structured manuscript which make it easy to follow and understand the story for readers. The manuscript intro synthesized the literatures appropriately.
Following please find my minor revisions and suggestions on the Abstract, Methods and Results:
Line 9-10: Please correct the line spacing.
Line 26: Suggestion: and as a promoter of growth and development of Pinus thunbergii seedlings.
line 97: (Figure 1. C)
Figure 2: Please indicate on the figure caption, the error bars are SD or SE?
Line 117: chitinase activity (A) and β -1,3-glucanase activity (B)
Line 147: Please suggest the temperature and relative humidity of the cold storage that the seeds were stored.
Line 148: running tap water?
Line 148: Please mention at what depth seeds were sown
Line 148: Suggestion: Please mention the dimension and trays size that were used in this study (based on figure 1 B) [Note: Pots in figure 1 A, B and C looks different]
Line 149: Suggestion: "Seedlings emerged in March 2014, approximately 4 months after planting date."
Line 149: You may delete "(Figure 1)"
Line 151: filled with how much media? Example: ... ''were manually filled with 250 mL substrate''.
Line 155: Temperature kept at 20-25 C day and night? Please mention the light regime of the greenhouse as well.
line 165-166: Please add a reference if any available? It would be informative for the audience/readers of the manuscript.
Line 172: How about the plant height? looking at the table 2 column 4th from the left, the value of the plant height = shoot length + root length? Are they measured from the same seedlings? If yes why the reported value for control and chemical fertilizer are slightly higher than shoot length (column2)+Root length (column 3)?
Results
Line 202: Must be Figure 3A
Line 204: Must be Figure 3B
Line 206: Must be Figure 3.
Left picture label "A. Control"
Line 222: Suggestion: Means within each column with the same letter...
Line 239: Please delete _ at the end of line 239
Table 2: FIND MY NOTE ON LINE 172. Control total height must be 28.89 cm
Chemical fertilizer total height must be 30.85 cm Please advise if I am missing something?
Line 243: Suggestion: Means within each column with the same letter...
Line 273: to Combat
Line 280: Must be Figure 4
Line 284: must be Figure 3B. please find my comments on page 6 Line 206.
Page 9: Need a label for the graph of pathogen growth inhibition %
Page 9: A. Control Or add labels on the pictures as (A) and (B)
Line 291: Must be Figure 4. Also please rewrite the caption since in the current form it is confusing!
The caption might be as follow: Growth inhibition percentage shown in graph X and ... shown in pictures (A) control and (B) ...
Line 316: You may replace height with "shoots"
Line 319: Please double check the citation of the figures in the text because of the confusion of figure numbering in the body of the manuscript as well as figures captions.
Line 328: Fund:
Line 334-338: Please double check the author guideline on Forest MDPI journal if it is necessary to use initials instead of full name in this section.

Author Response
Response to Reviewer 2 Comments
We appreciate your valuable time and critical comments. We agree with your suggestions and comments. Therefore, we have revised the manuscript according to your suggestions. The revised sentences are highlighted in yellow in the revised manuscript.
All errors in data in Tables 1 and 2 have been corrected. These errors did not affect the results of this study. In addition, the volume of container has been changed from 250 mL to 500 mL.
*********************
Great work. A well-structured manuscript which make it easy to follow and understand the story for readers. The manuscript intro synthesized the literatures appropriately.
Following please find my minor revisions and suggestions on the Abstract, Methods and Results:
- Line 9-10: Please correct the line spacing.
► We have corrected it. (Lines 9-11)
- Line 26: Suggestion: and as a promoter of growth and development of Pinus thunbergii seedlings.
► We have revised it. (Line 27)
- line 97: (Figure 1. C)
► We have revised it. (Line 96)
- Figure 2: Please indicate on the figure caption, the error bars are SD or SE?
► We have inserted “Error bars represent standard deviation of three replications” in Figure 2. (Line 117)
- Line 117: chitinase activity (A) and β -1,3-glucanase activity (B)
► We have revised it in Figure 2. (Line 116)
- Line 147: Please suggest the temperature and relative humidity of the cold storage that the seeds were stored.
► We have revised it as follows:
Coastal pine seeds used in this study were obtained in December 2013 from the National Forest Seed Variety Center of Korea and stored at 4°C in a cool refrigerator with constant relative humidity of 40%. (Lines 145-147)
- Line 148: running tap water?
► Yes. (Line 147)
- Line 148: Please mention at what depth seeds were sown
► We have inserted “Seeds were sown at 1 cm depth of growth media in each container (volume, 500 mL)”. (Line 147-148)
► In addition, the volume of container has been changed from 250 mL to 500 mL.
- Line 148: Suggestion: Please mention the dimension and trays size that were used in this study (based on figure 1 B) [Note: Pots in figure 1 A, B and C looks different]
► We have inserted the information about the dimension and trays size as follows:
The dimension of seedling tray was 25 cm wide ´ 41 cm long ´ 16 cm high, including 15 cells which was 7.5 cm diameter ´ 16 cm high (Figure 1B). (Lines 148-150)
- Line 149: Suggestion: "Seedlings emerged in March 2014, approximately 4 months after planting date."
► We have inserted to “Seedling emerged in March 2014, approximately 1 month after sowing seeds in February 2018”. (Lines 150-151)
- Line 149: You may delete "(Figure 1)"
► We have deleted it. (Line 150)
- Line 151: filled with how much media? Example: ... ''were manually filled with 250 mL substrate''.
► We have inserted to “---- were manually filled with approximately 500 mL substrate”. (Line 153)
- Line 155: Temperature kept at 20-25 C day and night? Please mention the light regime of the greenhouse as well.
► We have inserted “--- temperature kept at 20-25°C all day long with a heating and cooling system. The light condition was natural”. (Lines 157-158)
- Line 165-166: Please add a reference if any available? It would be informative for the audience/readers of the manuscript.
► We have added a reference in Kim et al. (2012). (Line 169 and Lines 471-473)
- Line 172: How about the plant height? looking at the table 2 column 4th from the left, the value of the plant height = shoot length + root length? Are they measured from the same seedlings? If yes why the reported value for control and chemical fertilizer are slightly higher than shoot length (column2)+Root length (column 3)?
► The data had error in the original Table 2. All these errors have been corrected. However, these errors did not affect the results of this study. (Line 246)
Results
- Line 202: Must be Figure 3A
► We have revised it. (Line 206)
- Line 204: Must be Figure 3B
► We have revised it. (Line 208)
- Line 206: Must be Figure 3.
► We have revised it. (Line 210)
- Left picture label "A. Control"
► We have revised it. (Line 209)
- Line 222: Suggestion: Means within each column with the same letter...
► We have revised it. (Line 226)
- Line 239: Please delete _ at the end of line 239
► We have deleted it. (Line 243)
- Table 2: FIND MY NOTE ON LINE 172. Control total height must be 28.89 cm
Chemical fertilizer total height must be 30.85 cm Please advise if I am missing something?
► The data had error in the original Table 2. All these errors have been corrected. However, these errors did not affect results of this study. (Line 246)
- Line 243: Suggestion: Means within each column with the same letter…
► We have revised it. (Line 247)
- Line 273: to Combat
► This sentence has been deleted. We have rewritten the Discussion section based on specific results of this study with reference to the literature according to suggestions of another reviewer. (Line 265)
- Line 280: Must be Figure 4
► We have revised it. (Line 273)
- Line 284: must be Figure 3B. please find my comments on page 6 Line 206.
► We have revised it. (Line 272)
- Page 9: Need a label for the graph of pathogen growth inhibition %
► We have inserted a label A to Figure 4. (Line 276)
- Page 9: A. Control Or add labels on the pictures as (A) and (B)
► We have inserted a label B including (a) and (b) to Figure 4. (Line 276)
- Line 291: Must be Figure 4. Also please rewrite the caption since in the current form it is confusing!
► We have revised it. (Line 277)
- The caption might be as follow: Growth inhibition percentage shown in graph X and ... shown in pictures (A) control and (B) ...
► We have revised it as follows:
Growth inhibition percentage shown in a graph (A) and antagonistic activity shown in pictures (B) for control (a) and B. licheniformis MH48 (b) against F. oxysporum by culture method. (Lines 277-278)
- Line 316: You may replace height with "shoots"
► We have replaced it with “shoots”. (Line 298)
- Line 319: Please double check the citation of the figures in the text because of the confusion of figure numbering in the body of the manuscript as well as figures captions.
► We have revised the citation of figures in the text. (Line 301)
- Line 328: Fund:
► We have revised it. (Line 310)
- Line 334-338: Please double check the author guideline on Forest MDPI journal if it is necessary to use initials instead of full name in this section.
► We have used initials instead of full name. (Lines 316-319)

Round 2
Reviewer 1 Report
Thank you for your response to my comments and the revisions to the text. This mainly satisfies my concerns although I have noticed a small mistake.
* in the revised caption for figure 2, the caption does not match the figure, which does not have error bars. Was the revised figure not uploaded?
Author Response
Response to Reviewer 1 Comments
We appreciate your valuable time and critical comments.
*********************
Thank you for your response to my comments and the revisions to the text. This mainly satisfies my concerns although I have noticed a small mistake.
* in the revised caption for figure 2, the caption does not match the figure, which does not have error bars. Was the revised figure not uploaded?
► We have indicated error bar for Figure 2 in previously submitted MS. However, the range of the error bars were so narrow that it was hidden from the solid circles. Corrected figure shows error bar.
